# Evaluation of a Novel Chromogenic Medium for the Detection of *Pseudomonas aeruginosa* in Respiratory Samples from Patients with Cystic Fibrosis

**DOI:** 10.3390/microorganisms10051004

**Published:** 2022-05-10

**Authors:** Thang V. Truong, Alexander Twist, Andrey Zaytsev, Emma C. L. Marrs, Audrey Perry, Graeme Turnbull, Sylvain Orenga, Stephen P. Stanforth, John D. Perry

**Affiliations:** 1Department of Applied Sciences, Northumbria University, Newcastle upon Tyne NE1 8ST, UK; viet.t.truong@northumbria.ac.uk (T.V.T.); alexander.twist@st-annes.ox.ac.uk (A.T.); a.v.zaytsev1@outlook.com (A.Z.); g.turnbull@northumbria.ac.uk (G.T.); steven.stanforth@northumbria.ac.uk (S.P.S.); 2Microbiology Department, Freeman Hospital, Newcastle upon Tyne NE7 7DN, UK; e.marrs@nhs.net (E.C.L.M.); audrey.perry@nhs.net (A.P.); 3Research & Development Microbiology, bioMérieux SA, 3 Route de Port Michaud, 38 390 La-Balme-les-Grottes, France; sylvain.orenga@biomerieux.com

**Keywords:** *Pseudomonas aeruginosa*, cystic fibrosis, culture media, chromogenic, aminopeptidase

## Abstract

*Pseudomonas aeruginosa* is a dominant cause of respiratory infection in individuals with cystic fibrosis (CF), leading to significant morbidity and mortality. Detection of *P. aeruginosa* is conducted by culture of respiratory samples but this process may occasionally be compromised due to overgrowth by other bacteria and fungi. We aimed to evaluate a novel chromogenic medium, Pseudomonas aeruginosa chromogenic agar (PACA), for culture of *P. aeruginosa* from respiratory samples, from patients with CF. A total of 198 respiratory samples were cultured onto PACA and three other media: CHROMID^®^
*P. aeruginosa*, CHROMagar™ Pseudomonas and MacConkey agar. *P. aeruginosa* was recovered from 66 samples (33%), using a combination of all media. After 72 h incubation, the sensitivity of the four chromogenic media was as follows: 91% for PACA and CHROMagar™ Pseudomonas, 85% for CHROMID^®^
*P. aeruginosa* and 83% for MacConkey agar. For the three chromogenic media, the positive predictive value after 72 h was as follows: 95% for PACA, 56% for CHROMagar™ Pseudomonas and 86% for CHROMID^®^
*P. aeruginosa*. PACA proved to be a highly effective culture medium for the isolation and specific detection of *P. aeruginosa* from respiratory samples.

## 1. Introduction

*Pseudomonas aeruginosa* (PSA) is the most dominant cause of respiratory infection in individuals with cystic fibrosis (CF) and its prevalence increases with age. According to data from the United Kingdom CF Registry, chronic infection with *P. aeruginosa* affects 5.2% of those <16 years of age and 39.4% of those >16 years [1]. Once *P. aeruginosa* becomes adapted to the lung environment of individuals with CF, it can be extremely difficult to eradicate, leading to long-term morbidity and mortality [2,3]. In order to prevent, or at least postpone, the establishment of chronic infection, eradication programs have been widely adopted using early and aggressive treatment regimens when *P. aeruginosa* is first detected [4]. It is, therefore, important to employ sensitive laboratory methods to detect *P. aeruginosa* in respiratory samples, including oropharyngeal swabs or ‘cough swabs’ that are frequently used in young children who may be unable to produce sputum [5,6].

In most clinical laboratories, the detection of *P. aeruginosa* relies upon culture, which remains the ‘gold standard’ method [7]. In the UK, national laboratory standards recommend the inclusion of cystine–lactose–electrolyte-deficient (CLED) agar or MacConkey agar to improve the recovery of *P. aeruginosa* from individuals with CF [8]. The use of MacConkey agar is supported by data from Doern et al., who compared a range of non-selective and selective media for the isolation of respiratory tract pathogens from individuals with CF. From 253 samples, they recovered *P. aeruginosa* from 137 samples and all of these isolates were recovered on MacConkey agar (100% sensitivity) [9]. Laine et al. evaluated a novel chromogenic agar (PS-ID; later commercialized as CHROMID^®^
*P. aeruginosa*) for the detection of *P. aeruginosa* from sputum samples and compared it with culture on blood agar and two other selective agars [10]. They demonstrated that PS-ID offered a presumptive identification of *P. aeruginosa,* with high predictive value (98.3%) but only a marginal improvement in sensitivity (95.2%), when compared with non-selective blood agar (90.3%).

The aim of this study was to evaluate a new chromogenic agar medium (*P. aeruginosa* chromogenic agar; PACA) for isolation of *P. aeruginosa* from respiratory samples from individuals with CF, which utilizes a novel enzyme substrate for the detection of β-alanyl aminopeptidase. The medium was evaluated with 198 respiratory samples and compared with three other culture media: MacConkey agar, CHROMID^®^
*P. aeruginosa* and CHROMagar™ Pseudomonas. 

## 2. Materials and Methods

### 2.1. Culture Media

CHROMID^®^
*P. aeruginosa* (Ref: 43462) and MacConkey agar (Ref: 43141) were supplied as pre-poured plates from bioMérieux, Craponne, France. CHROMagar™ Pseudomonas (Ref: PS832) was obtained from CHROMagar, Paris, France, as a dehydrated powder. For preparation, 22.75 g of powder was suspended in 500 mL deionized water and heated at 100 °C for approximately 5 min. The molten medium was poured into sterile Petri dishes at 50 °C and allowed to set. Plates were then dried in a hot room at 37 °C for five minutes. Plates were prepared weekly as required. 

The basal medium used for PACA was Columbia agar kindly supplied by bioMeriéux, Craponne, France. Other ingredients included cellobiose and N-methyl-2-pyrolidone (both from Merck, Gillingham, UK), magenta-β-glucoside (Glycosynth, Warrington, UK) and C-390 (9-chloro-9-[4-(diethylamino) phenyl]-9,10-dihydro-10-phenylacridine hydrochloride), which was synthesized at Northumbria University (Newcastle upon Tyne, UK) but the compound is also available commercially from Biosynth (Staad, Switzerland). Finally, a novel chromogenic substrate β-alanyl-ANDF (N-(β-alanyl)-2-amino-7-nitro-9,9-dimethylfluorene TFA salt) was synthesized at Northumbria University. The structure and method for synthesis of this novel substrate are provided as Appendix A. A single batch of substrate was used for the entire study.

PACA was prepared in house as follows: 45.8 g of Columbia agar, 0.1 g cellobiose and 0.004 g C-390 were added to 1 L of deionized water and heated at 100 °C for approximately 5 min. After cooling to 50 °C in a water bath, two chromogenic substrates were added (each dissolved in 1 mL of N-methyl-2-pyrolidone): β-alanyl-ANDF (50 mg) and magenta-β-glucoside (75 mg). After mixing, the molten medium was poured into sterile Petri dishes at 50 °C and allowed to set. Plates were then dried in a hot room at 37 °C for five minutes. Plates were prepared every two weeks as required.

### 2.2. Quality Control of Culture Media

Strains used for quality control of culture media were obtained from the National Collection of Type Cultures (NCTC), Colindale (London, UK), UK. Each batch of media was challenged with approx. 10^5^ colony-forming units (CFU) of the following strains: *P. aeruginosa* NCTC 12903, *Escherichia coli* NCTC 12241, *Enterococcus faecalis* NCTC 12697 and *Staphylococcus aureus* NCTC 12973. This was achieved by using 1 µL of a 0.5 McFarland suspension prepared using a Densimat (bioMérieux, Basingstoke, UK), from a fresh overnight culture on Columbia blood agar. After 48 h incubation at 37 °C, only *P. aeruginosa* and *E. coli* were able to grow on MacConkey agar whereas only *P. aeruginosa* was able to grow on the three chromogenic media. 

### 2.3. Culture of Clinical Samples

Over a two-month period, respiratory samples were referred to the Freeman Hospital Microbiology Department from 198 distinct patients who were attending specialized clinics for individuals with CF. The samples included 131 cough swabs and 67 sputum samples. The specimens were collected as part of routine monitoring of the patients and no specimens were collected for the purposes of this study. As part of routine processing, sputum samples were treated with an equal volume of sputasol (Product SR0233A; Oxoid, Basingstoke, UK) and mixed thoroughly using a vortex mixer until homogeneous. All respiratory samples were then cultured for a range of pathogens in line with routine laboratory procedures. Leftover aliquots of these samples were then anonymized by laboratory staff and processed as described below.

Material from each cough swab was suspended in 1 mL of sterile physiological saline (0.85%) and, after vortexing for 30 s, a 50 µL aliquot was inoculated onto each of the four test media. For homogenized sputum samples, a 50 µL aliquot was inoculated onto each of the same four media. The inocula on the four culture media were spread to obtain isolated colonies and all media were incubated at 37 °C for 72 h. All four media were examined and colonies investigated after 24, 48 and 72 h (see Section 2.4). 

### 2.4. Identification

All bacteria and yeasts recovered on all media were subcultured onto Columbia blood agar, incubated overnight at 37 °C, and subsequently identified using matrix-assisted laser desorption/ionization-time-of-flight mass spectrometry (MALDI-TOF MS) in accordance with manufacturer’s instructions (Bruker, Coventry, UK). A bacterial test standard supplied by the manufacturer was used for daily calibration of the instrument and a collection of NCTC control strains representing 14 different bacteria and yeasts were tested weekly as part of routine quality control, consistent with recommendations in UK national standards [11].

### 2.5. Interpretation of Chromogenic Culture Media

For colonies recovered on chromogenic media, the following were regarded as presumptive *P. aeruginosa*: (i) yellow or yellow-green colonies on PACA; (ii) green or green-blue colonies on CHROMagar™ Pseudomonas; (iii) red, pink or purple colonies on CHROMID^®^
*P. aeruginosa*; (iv) colonies on any medium producing a green or brown diffusible pigment.

### 2.6. Statistical Analysis

The sensitivity and the rate of false positives for each culture medium was compared with each other medium using McNemar’s test with the continuity correction applied. 

## 3. Results

### 3.1. Detection of P. aeruginosa from Clinical Samples

Using a combination of all media, *P. aeruginosa* was recovered from 66 samples (exactly one-third of all samples); 38 sputum samples and 28 cough swabs. On the basis that typical chromogenic reactions are required for successful detection of *P. aeruginosa* on chromogenic media, the most positive samples were detected by CHROMagar™ Pseudomonas and PACA (both with a sensitivity of 91%), followed by CHROMID^®^
*P. aeruginosa,* with a sensitivity of 85%. MacConkey agar recovered *P. aeruginosa* from 83% of positive samples but did not allow for differentiation from most other flora. There was no statistical difference between any of the media for recovery of *P. aeruginosa* (*p* > 0.05). The ‘time to detection’ for CHROMagar™ Pseudomonas and PACA was almost identical (see Figure 1).

Figure 2 shows the positive predictive value (PPV) of the three chromogenic media. The PPV was high for both CHROMID^®^
*P. aeruginosa* and PACA over the 72 h incubation period (with no statistical difference), whereas the PPV for CHROMagar™ Pseudomonas was substantially lower. On CHROMagar™ Pseudomonas, the number of samples from which false positive isolates were recovered was higher when compared with either CHROMID^®^
*P. aeruginosa* or PACA, and this difference was statistically significant at any timepoint (*p* ≤ 0.0001). Species that frequently produced green colonies indistinguishable from *P. aeruginosa* on CHROMagar™ Pseudomonas included *Achromobacter* species (*n* = 13), *Stenotrophomonas maltophilia* (*n* = 14) and other species of *Pseudomonas* (*n* = 14). Table 1 shows the species that produced ‘false-positive’ colonies on each of the three chromogenic media. PACA was the most specific medium, with only six isolates generating colonies resembling *P. aeruginosa* over the 72 h incubation period, compared to nine on CHROMID^®^
*P. aeruginosa*.

Figure 3 shows the appearance of *P. aeruginosa* on the four test media. Forty-seven percent of *P. aeruginosa* isolates recovered on CHROMagar™ Pseudomonas and MacConkey agar demonstrated a mucoid phenotype, compared with 37% of isolates recovered on PACA and 32% of isolates recovered on CHROMID^®^
*P. aeruginosa*. 

### 3.2. Selectivity of the Four Test Media

Species other than *P. aeruginosa* were frequently isolated on all media. PACA was the most selective of the four media, with only 51 other isolates recovered. This compared with 63 on CHROMagar™ Pseudomonas, 74 on CHROMID^®^
*P. aeruginosa* and 91 on MacConkey agar. All media were generally highly effective at inhibiting yeasts and fungi, which are common in these samples. Amongst the three chromogenic media, PACA was better at inhibiting non-fermentative Gram-negative bacteria, such as *S. maltophilia* and other *Pseudomonas,* but less effective at inhibiting Enterobacterales (see Table 2).

## 4. Discussion

We describe a new chromogenic medium for the isolation and presumptive identification of *P. aeruginosa* from respiratory samples of individuals with CF. The medium employs a novel substrate that is hydrolyzed by β-alanyl aminopeptidase to generate a yellow dye that is also fluorescent. The dye remains highly restricted to bacterial colonies. *P. aeruginosa* is known to produce this enzyme, along with only a few other Gram-negative species, such as *Serratia* species and *Burkholderia cepacia* complex [12]. The only other chromogenic medium specifically designed for *P. aeruginosa* is CHROMID^®^
*P. aeruginosa*, which also utilizes a chromogenic substrate, β-alanyl pentylresorufamine, for the detection of this enzyme [13]. The published method for the preparation of β-alanyl pentylresorufamine [14], which is currently utilized in CHROMID^®^
*P. aeruginosa*, has several drawbacks. These include numerous synthetic steps and purification problems with reactions that produce mixtures of products. The synthesis of the fluorene-based substrate described in this paper (see Appendix A) overcomes these problems and uses a three-step sequence commencing from commercially available and relatively inexpensive starting materials. Other ingredients of PACA include C-390, which is an effective selective agent for the inhibition of flora, other than *P. aeruginosa* [15], and a chromogenic substrate for β-glucosidase (magenta-β-glucoside) to encourage *Serratia* species to produce red colonies, thus, differentiating them from *P. aeruginosa* after 48–72 h incubation.

PACA proved to be more selective than the other three test media and its high selectivity means that it does not need to be autoclaved at temperatures above 100 °C. Its sensitivity was as high as that of any other test medium, with a superior positive predictive value. After 72 h incubation, only 3 isolates generated false-positive colonies on PACA, compared with 47 on CHROMagar™ Pseudomonas and 9 on CHROMID^®^
*P. aeruginosa*. CHROMagar™ Pseudomonas is designed for the isolation of *Pseudomonas* species from environmental samples and does not claim to differentiate *P. aeruginosa*. This medium proved to be effective at isolating *P. aeruginosa* from clinical samples. However, in the context of CF, it lacked specificity, with a range of non-fermentative Gram-negative species also resembling *P. aeruginosa*.

The use of MacConkey (or CLED) agar is recommended by national laboratory standards in the UK to enhance the isolation of *P. aeruginosa* from individuals with CF [8]. In this study, the sensitivity of MacConkey agar was limited to 83% (55 positive samples), even when every distinct isolate (*n* = 146) was investigated using MALDI-TOF mass spectrometry. The absence of lactose acidification is of limited value to distinguish *P. aeruginosa* on MacConkey agar, as many isolates were recorded as producing pink colonies after 72 h incubation. The use of PACA enabled the recovery of *P. aeruginosa* from five additional samples (sensitivity: 91%) and would have required the investigation of only 63 isolates in total (i.e., those producing yellow or yellow-green colonies on PACA) if cultures were read after 72 h incubation. Thus, the use of a specific chromogenic agar has the potential to reduce the amount of labor time (and materials) required for processing colonies.

A limitation of using PACA is that the typical yellow coloration of *P. aeruginosa* can be obscured by isolates with strong natural pigmentation. However, such isolates should always be investigated as potential isolates of *P. aeruginosa*. Furthermore, we have only evaluated the medium with samples from patients with CF and further validation would be required before applying the medium to samples from other patient groups or environmental samples. It is known, for example, that the activity of C-390, one of the key selective agents utilized in PACA, can be compromised by the use of certain types of membrane filters often used for environmental testing [16]. A final limitation of the study is that each batch of PACA was prepared using a single batch of the novel chromogenic substrate β-alanyl-ANDF. Further work is required to validate the consistent performance of different batches of substrate.

## Figures and Tables

**Figure 1 microorganisms-10-01004-f001:**
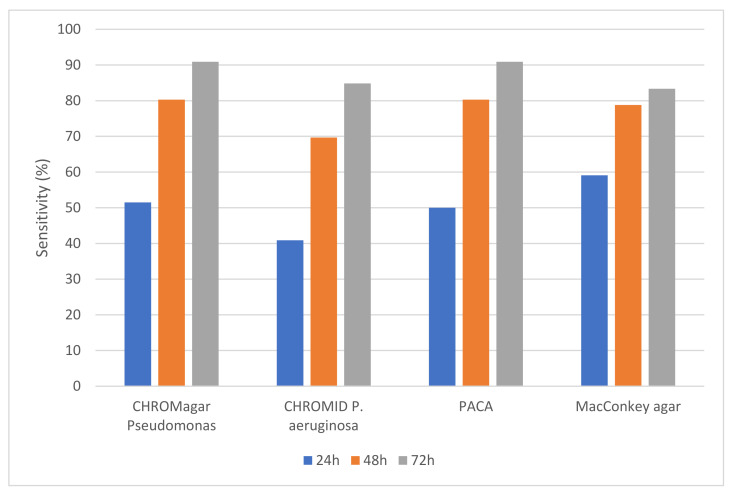
Sensitivity (%) of the four culture media for detection of *P. aeruginosa* from respiratory samples. Detection on the three chromogenic media depended on the generation of colored colonies (as detailed in Section 2.5).

**Figure 2 microorganisms-10-01004-f002:**
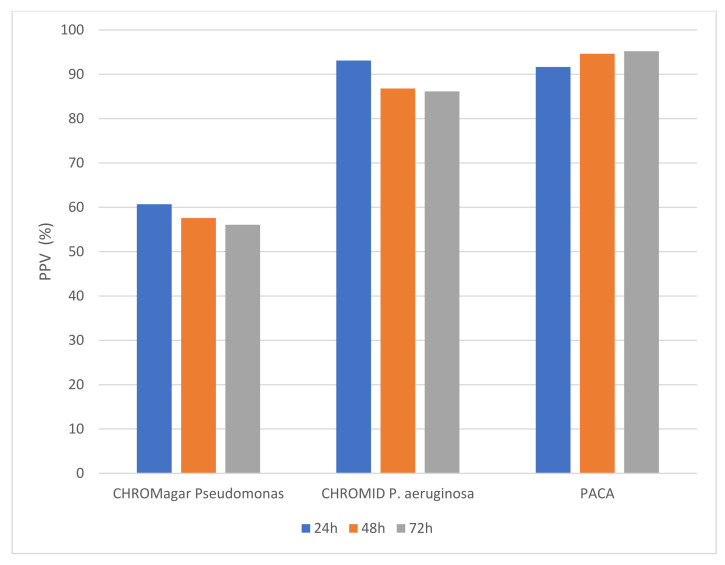
Positive predictive value (%) for colony color generated on three chromogenic media.

**Figure 3 microorganisms-10-01004-f003:**
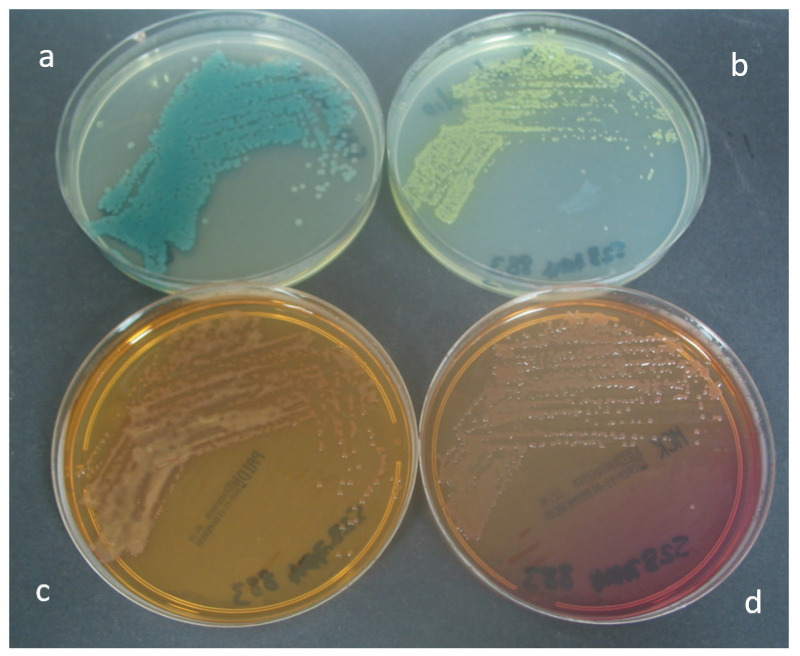
Appearance of *P. aeruginosa* after 24 h incubation on (**a**) CHROMagar™ Pseudomonas, (**b**) PACA, (**c**) CHROMID^®^
*P. aeruginosa* and (**d**) MacConkey agar.

**Table 1 microorganisms-10-01004-t001:** Species incorrectly assigned as presumptive *P. aeruginosa* on three chromogenic media.

	CHROMagar™ Pseudomonas			CHROMID^®^ *P. aeruginosa*			PACA		
	24 h	48 h	72 h	24 h	48 h	72 h	24 h	48 h	72 h
*Achromobacter* species	2	9	13	0	0	0	0	0	0
*Acinetobacter* species	1	1	1	0	0	0	0	0	0
*Burkholderia multivorans*	0	0	1	0	2	2	0	1	1
*Candida albicans*	0	0	1	0	0	0	0	0	0
*Candida parapsilosis*	0	0	1	0	0	0	0	0	0
*Escherichia coli*	1	1	1	0	0	0	0	0	0
*Ochrobactrum anthropi*	0	0	0	0	0	1	0	0	0
*Pseudomonas stutzeri*	0	1	1	0	0	0	0	0	0
*Pseudomonas fluorescens* group	3	5	5	0	3	3	0	0	0
*Pseudomonas oryzihabitans*	2	2	2	0	0	0	0	0	1
*Pseudomonas putida* group	2	5	5	0	0	0	0	0	0
Other *Pseudomonas* species	1	1	1	1	1	1	0	0	0
*Serratia* species	1	1	1	1	1	2	3	1	0
*Stenotrophomonas maltophilia*	9	13	14	0	0	0	0	1	1
Total false positives	22	39	47	2	7	9	3	3	3
Positive predictive value (%)	61	58	56	93	87	86	92	95	95

**Table 2 microorganisms-10-01004-t002:** Species other than *P. aeruginosa* recovered on the four test media during the 72 h incubation.

	CHROMagar™ Pseudomonas	CHROMID^®^ *P. aeruginosa*	PACA	MacConkey Agar
No growth	91	82	93	68
*Achromobacter* species	18	19	14	16
*Acinetobacter* species	1	1	2	2
*Bordetella* species	0	1	0	0
*Burkholderia multivorans*	1	2	1	1
*Candida albicans*	1	0	0	0
*Candida parapsilosis*	1	1	0	5
*Enterobacter* species	3	4	7	8
*Enterococcus faecalis*	0	0	0	3
*Escherichia coli*	3	2	0	6
*Klebsiella aerogenes*	1	1	0	0
*Klebsiella oxytoca*	0	0	4	5
*Klebsiella pneumoniae*	0	0	0	1
*Klebsiella variicola*	0	2	0	2
*Morganella morganii*	0	0	0	1
*Neisseria subflava*	0	1	0	0
*Ochrobactrum anthropi*	0	1	0	1
*Pantoea agglomerans*	0	0	0	2
*Proteus mirabilis*	1	1	0	3
*Providencia rettgeri*	0	0	1	0
*Pseudescherichia vulneris*	0	0	0	1
*Pseudomonas fluorescens* group	5	4	3	3
*Pseudomonas oryzihabitans*	2	2	1	1
*Pseudomonas putida* group	5	4	3	2
*Pseudomonas stutzeri*	1	1	0	1
Other *Pseudomonas* species	2	2	0	0
*Ralstonia mannitolilytica*	0	1	0	1
*Rothia mucilaginosa*	0	1	0	0
*Serratia* species	3	4	7	10
*Staphylococcus aureus*	0	3	0	0
*Stenotrophomonas maltophilia*	15	16	8	14
*Streptococcus oralis*	0	0	0	1
*Streptococcus sanguinis*	0	0	0	1
**Total**	63	74	51	91

## Data Availability

All relevant data are included in the manuscript.

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
