# Peer review of "Evaluation of a Novel Chromogenic Medium for the Detection of Pseudomonas aeruginosa in Respiratory Samples from Patients with Cystic Fibrosis"

_microorganisms, 2022, doi:10.3390/microorganisms10051004_

Round 1

Reviewer 1 Report

The manuscript by Troung et al. describes a new medium suitable for testing of P. aeruginosa strains as a selective and specific method to identify the pathogen. The work was performed on pathogen samples from CF patients and the statistics are impressive. The manuscript has 2 flaws:

  1. The names of pathogens are frequently not italicized. The correction is simple.
  2. CF patients are known to have rearrangements in the genome leading to the deactivation of T3SS and other virulence components. I have not seen a comparison with normal samples, not collected from CF patients, to validate the system. Would it be possible to run a test showing that human samples from CF patients do not compromise the detection method?

Author Response

Responses to reviewer 1:

The manuscript by Troung et al. describes a new medium suitable for testing of P. aeruginosa strains as a selective and specific method to identify the pathogen. The work was performed on pathogen samples from CF patients and the statistics are impressive. The manuscript has 2 flaws:

  1. The names of pathogens are frequently not italicized. The correction is simple.

Author response: Thank you for your comments. We have ensured consistent use of italics. The only exception is when bacterial names form part of the name of a commercial product, for example, “CHROMID P. aeruginosa” or “CHROMagar Pseudomonas”. In such instances we have used the formatting that is specified by the manufacturer. We believe this is the most appropriate approach and it is also consistent with previous publications. For consistency, the same approach has been applied to the experimental medium “Pseudomonas aeruginosa chromogenic agar”.

  1. CF patients are known to have rearrangements in the genome leading to the deactivation of T3SS and other virulence components. I have not seen a comparison with normal samples, not collected from CF patients, to validate the system. Would it be possible to run a test showing that human samples from CF patients do not compromise the detection method?

Author response: With respect to the last sentence, we have shown clearly that samples from patients with CF do not compromise detection, as we only used samples from CF. Furthermore, in our quality control procedures we demonstrate that P. aeruginosa NCTC 12903 (not derived from CF) demonstrates typical colonies. It would be possible to culture a series of sputum samples from patients who do not have underlying lung disease (such as that caused by cystic fibrosis), however the yield of P. aeruginosa is very small from such samples. Consequently, it would likely be necessary to culture many hundreds of such samples to obtain meaningful results. Despite this, we take the reviewer’s point that the medium cannot be assumed to work well in scenarios outside of this setting and we are aware of limitations that might occur when testing, for example, environmental samples. Text has therefore been added to line 242 to state that “…..further validation would be required before applying the medium to samples from other patient groups or environmental samples”. An additional reference is also provided.

Reviewer 2 Report

Manuscript microorganisms-1668437 by Truong et al is an original study describing the development and testing of a new chromogenic culture medium allowing detection of Pseudomonas aeruginosa in samples from patients with cystic fibrosis (CF).  Respiratory infections with P. aeruginosa contribute to considerable morbidity and mortality in CF patients.  Early pathogen detection is essential to facilitate initiation of treatment regimens.  In the present study, the authors compare a new selective and differential medium, PACA to other existing media for the ability to detect and discriminate P. aeruginosa in CF patient samples.  The study concludes that PACA is as efficient or better than other media at providing a presumptive P. aeruginosa identification.  Overall, the study is well-written and interesting.  However, there are several points that require clarification:

1) The authors state that media batches were routinely quality control tested.  However, critical information regarding the novel beta-alanyl-ANDF substrate used in the study is missing.  Based on the description provided, it is unclear as to whether a single batch of b-alanyl-ANDF was synthesized and used throughout the study or whether multiple batches were synthesized on different days and each tested as part of the study.  Thus, it is unclear as to whether synthesis of the substrate is reproducible from batch to batch and how different substrate batches perform in the assay/isolation medium.  This raises questions regarding the overall potential utility of the medium. The manuscript states that information regarding the novel substrate used in the PACA medium is provided as supplementary information.  However, this reviewer was unable to find or access this material.  The authors should provide information regarding b-alanyl-ANDF batch numbers tested in the main text and update their supplemental material as needed.  If only one batch of b-alanyl-ANDF was synthesized and tested, then the authors must mention this as a potential limitation of their study.

2) The authors state in the discussion that PACA PPV performance was superior, however, the only data where a statistical analysis is mentioned is for the sensitivity comparison.  Thus, it is unclear as to whether PACA PPV is statistically superior to the other differential/selective media tested in the study.  The authors should substantiate this claim, or temper if the differences in PPV are not significant.

3) The authors cite a previous study where the sensitivity and PPV of CHROMID P. aeruginosa agar were reported as 95.2% and 98.3% respectively.  However, the authors do not discuss further how these published data compare to their own.  This would be useful to the reader and may provide further support for the performance of PACA and its potential utility as a novel differential/selective medium for P. aeruginosa identification.

4) Figures 1 and 2 use a bar graph to report a single data point for each condition shown.  The use of this type of a graph might be more compelling if the sensitivity and PPV data are shown for each batch of media tested (if possible).  Otherwise, these single data points might be better reported as a Table.  If the authors keep the bar graph format, they should add a more descriptive title to the y-axis rather than just using a % symbol so the figure stands alone.

5) There are several instances throughout the manuscript where P. aeruginosa is not in italic font.  This should be corrected.

Author Response

Responses to reviewer 2:

Manuscript microorganisms-1668437 by Truong et al is an original study describing the development and testing of a new chromogenic culture medium allowing detection of Pseudomonas aeruginosa in samples from patients with cystic fibrosis (CF).  Respiratory infections with P. aeruginosa contribute to considerable morbidity and mortality in CF patients.  Early pathogen detection is essential to facilitate initiation of treatment regimens.  In the present study, the authors compare a new selective and differential medium, PACA to other existing media for the ability to detect and discriminate P. aeruginosa in CF patient samples.  The study concludes that PACA is as efficient or better than other media at providing a presumptive P. aeruginosa identification.  Overall, the study is well-written and interesting.  However, there are several points that require clarification:

  • The authors state that media batches were routinely quality control tested.  However, critical information regarding the novel beta-alanyl-ANDF substrate used in the study is missing.  Based on the description provided, it is unclear as to whether a single batch of b-alanyl-ANDF was synthesized and used throughout the study or whether multiple batches were synthesized on different days and each tested as part of the study.  Thus, it is unclear as to whether synthesis of the substrate is reproducible from batch to batch and how different substrate batches perform in the assay/isolation medium.  This raises questions regarding the overall potential utility of the medium. The manuscript states that information regarding the novel substrate used in the PACA medium is provided as supplementary information.  However, this reviewer was unable to find or access this material.  The authors should provide information regarding b-alanyl-ANDF batch numbers tested in the main text and update their supplemental material as needed.  If only one batch of b-alanyl-ANDF was synthesized and tested, then the authors must mention this as a potential limitation of their study.

Author response: Thank you for your review and comments. The supplementary data was uploaded and should have been available to reviewers. I have ensured that it has been added to the main document in our revision so it is available for review. We have clarified in line 82 that a single batch of substrate was used for the entire study. As recommended, in line 247, we have highlighted this as a possible limitation by adding the following text: “A final limitation of the study is that each batch of PACA was prepared using a single batch of the novel chromogenic substrate ß-alanyl-ANDF. Further work is required to validate consistent performance of different batches of substrate”.

  • The authors state in the discussion that PACA PPV performance was superior, however, the only data where a statistical analysis is mentioned is for the sensitivity comparison.  Thus, it is unclear as to whether PACA PPV is statistically superior to the other differential/selective media tested in the study.  The authors should substantiate this claim, or temper if the differences in PPV are not significant.

Author response: As stated in line 154, the PPV was high for both CHROMID® P. aeruginosa and PACA over the 72h incubation period and we have now clarified that there was no statistical difference between these two media. However, the number of false positive colonies was much higher on CHROMagar Pseudomonas (p < 0.001) and this statistical difference has now been highlighted. The text in lines 154 – 160 has been revised accordingly.

3) The authors cite a previous study where the sensitivity and PPV of CHROMID P. aeruginosa agar were reported as 95.2% and 98.3% respectively.  However, the authors do not discuss further how these published data compare to their own.  This would be useful to the reader and may provide further support for the performance of PACA and its potential utility as a novel differential/selective medium for P. aeruginosa identification.

Author response: In the introduction we were careful to ensure that relevant previous studies were cited and the figures for PS-ID (a prototype of CHROMID P. aeruginosa) were cited. There is some difficulty in comparing the current results for CHROMID P. aeruginosa with those reported for PS-ID reported previously for two reasons:

  • It is generally difficult to compare different studies as the sensitivity is heavily influenced by the chosen comparators. In the study of Laine et al, one of the comparators was blood agar – which has no specific mechanism (or selectivity) for detection of P. aeruginosa. No other chromogenic medium was used as a comparator. It is therefore not surprising that the overall sensitivity of PS-ID proved to be high (95.2%) as it was up against relatively poor comparators.
  • A more significant problem is that Laine et al. did not test CHROMID P. aeruginosa but rather a prototype version (PS-ID). Alterations to the formulation frequently occur as part of the commercial larger-scale production process.

Due to a combination of these reasons, we believe it is preferable to resist making comparisons between the data reported here for the actual commercial product (CHROMID P. aeruginosa) and earlier data for a prototype version that may have a slightly different formulation.

4) Figures 1 and 2 use a bar graph to report a single data point for each condition shown.  The use of this type of a graph might be more compelling if the sensitivity and PPV data are shown for each batch of media tested (if possible).  Otherwise, these single data points might be better reported as a Table.  If the authors keep the bar graph format, they should add a more descriptive title to the y-axis rather than just using a % symbol so the figure stands alone.

Author response: As the reviewer has noted, these data could easily be tabulated. However, we believe the graph offers a “reader-friendly” visual representation of the data – for example showing the clear trend of increased sensitivity on each medium over the course of the incubation period. We have adjusted the Y-axis labels to be more descriptive as suggested.

It would not be practical to show the sensitivity for each batch of medium as some of the batches were very small. For example, CHROMagar was prepared as a 500 mL batch (as recommended by the manufacturer) thereby generating only 25 plates (fewer after quality control). We believe it would lead to confusion if we attempted to quote sensitivity and PPV for numerous small batches.

5) There are several instances throughout the manuscript where P. aeruginosa is not in italic font.  This should be corrected.

Author response: Thank you for highlighting this. We have now ensured consistent use of italics. The only exception is when bacterial names form part of the name of a commercial product, for example, “CHROMID P. aeruginosa” or “CHROMagar Pseudomonas”. In such instances we have used the formatting that is specified by the manufacturer. We believe this is the most appropriate approach and it is also consistent with previous publications. For consistency, the same approach has been applied to the experimental medium “Pseudomonas aeruginosa chromogenic agar”.

Reviewer 3 Report

Dear all,

the current review is presenting a novel method to identify Pseudomonas spp.

I donot have any negative comment except for the photo. I t should be of a higher quality. Otherwise, the manuscript is of a high quality. I agree to publish in the current version after editing the photo.

Kind regards and many thanks

Author Response

Responses to reviewer 3:

Dear all,

the current review is presenting a novel method to identify Pseudomonas spp.

I do not have any negative comment except for the photo. I t should be of a higher quality. Otherwise, the manuscript is of a high quality. I agree to publish in the current version after editing the photo.

Author response: Thank you for your review and positive comments. We used one of our better photographs (which may reflect poorly on my photography skills). I have replaced the photograph with one of higher quality. I hope it is acceptable to the journal production team.

Round 2

Reviewer 1 Report

A very well written and presented data. All questions have been answered.